# Correlation of High-Sensitivity Cardiac Troponin I Values and Cardiac Radiation Doses in Patients with Left-Sided Breast Cancer Undergoing Hypofractionated Adjuvant Radiotherapy with Concurrent Anti-HER2 Therapy

**Katarina Antunac [1,\*], Ljiljana Mayer [2], Marija Banovic [3] and Lidija Beketic-Oreskovic [1,4]**

[1] Division of Oncology and Radiotherapy, University Hospital for Tumours, Sestre Milosrdnice University Hospital Centre, Ilica 197, 10000 Zagreb, Croatia; lidijabeketicoreskovic@gmail.com

[2] Department of Medical Biochemistry in Oncology, University Hospital for Tumours, Sestre Milosrdnice University Hospital Centre, Ilica 197, 10000 Zagreb, Croatia; ljiljana.mayer@kbcsm.hr

[3] Polyclinic Leptir, Ilica 253, 10000 Zagreb, Croatia; mabanovi@gmail.com

[4] Department of Clinical Oncology, School of Medicine University of Zagreb, Salata 3 B, 10000 Zagreb, Croatia

\* Correspondence: katarina.antunac@gmail.com

**Abstract:** Anti HER2 therapy and left breast adjuvant radiation therapy (RT) can both result in cardiotoxicity. The aim of this study was to evaluate the influence of radiation dose on cardiac structures on the values of the early cardiotoxicity marker high-sensitivity cardiac troponin I (hscTnI) in patients with HER2-positive left breast cancer undergoing adjuvant concomitant anti-HER2 therapy and radiotherapy, and to establish a correlation between the hscTnI values and cardiac radiation doses. Sixty-one patients underwent left breast hypofractionated radiotherapy in parallel with anti-HER2 therapy: trastuzumab, combined trastuzumab–pertuzumab or trastuzumab emtansine (T-DM1). The hscTnI values were measured prior to and upon completion of radiotherapy. A significant increase in hscTnI was defined as >30% from baseline, with the second value being 4 ng/L or higher. Dose volume histograms (DVH) were generated for the heart, left ventricle (LV) and left anterior descending artery (LAD). The hscTnI levels were corelated with radiation doses on cardiac structures. An increase in hscTnI values was observed in 17 patients (Group 1). These patients had significantly higher mean radiation doses for the heart ($p = 0.02$), LV ($p = 0.03$) and LAD ($p = 0.04$), and AUC for heart and LV ($p = 0.01$), than patients without hscTnI increase (Group 2). The patients in Group 1 also had larger volumes of heart and LV receiving 2 Gy ($p = 0.01$ for both) and 4 Gy ($p = 0.02$ for both). LAD differences were observed in volumes receiving 2 Gy ($p = 0.03$), 4 Gy ($p = 0.02$) and 5 Gy ($p = 0.02$). The increase in hscTnI observed in patients receiving anti-HER2 therapy after adjuvant RT was positively associated with radiation doses on the heart, LV and LAD.

**Keywords:** HER2-positive breast cancer; adjuvant radiotherapy; radiotherapy dose hypofractionation; cardiotoxicity; high-sensitivity cardiac troponin I; trastuzumab; pertuzumab; trastuzumab emtansine

## 1. Introduction

Breast cancer adjuvant radiotherapy reduces the risk of disease recurrence (from 35% to 19.3% in 10 years) and the risk of breast cancer death (from 25.2% to 21.4% in 15 years) [1]. However, the incidental exposure of the heart during left breast irradiation increases the risk of ischemic heart disease that starts about 5 years after radiotherapy and continues up to the third decade upon its completion. The ischemic heart disease rate is proportional to the mean dose to the whole heart; it increases linearly by 7.4% per Gray

(Gy) and no threshold has been defined [2]. The risk of radiation-induced heart disease is not only dose-dependent but also correlates with the volume of the cardiac structure receiving a certain radiation dose [3]. Therefore, it was shown that the volume of the left ventricle (LV) receiving 5 Gy was the most important prognostic dose-volume parameter for the development of an acute coronary event [4]. A mean dose on the left anterior descending (LAD) artery higher than 5 Gy was associated with an increased requirement for coronary intervention in LAD [5].

Troponin is a contractile apparatus component in both cardiac and skeletal myocytes. Troponin I and T isoforms are highly specific to cardiac myocytes; their detection in serum is a specific marker of cardiac damage [6]. The prognostic value of small changes in high sensitivity cardiac troponins, those below the 99th centile, has been shown in diseases affecting the cardiac muscle, such as coronary disease [7]. Radiation-induced cardiac cell damage could be a consequence of changes in cardiac vasculature and inflammation caused by radiation or could be the direct effect of radiation on the destruction of myocyte membranous structures [8,9].

A meta-analysis of eight randomised trials evaluated whether high-sensitivity cardiac troponin T (hscTnT) can be used as an early diagnostic marker of cancer-treatment-related cardiac dysfunction. A correlation between elevated hscTnT levels and cancer-treatment-related cardiac dysfunction was found and hscTnT testing improved the accuracy of the diagnosis. Still, it was not possible to define exact cut-off values of hscTnT for the early diagnosis of cardiac dysfunction [10].

A correlation between radiation doses on cardiac structures and the increase in troponin I levels during and after radiotherapy has been shown in patients undergoing adjuvant irradiation of the left breast. The patients in this trial did not receive chemotherapy or anti-HER2 therapy [11].

Patients with early HER2-positive breast cancer receive anti-HER2 systemic therapy based on trastuzumab in both neoadjuvant and adjuvant settings. The overall treatment time is usually 1 year and adjuvant radiotherapy of the breast cancer is most often applied concomitantly with anti-HER2 therapy. The most common side effect of trastuzumab is cardiotoxicity, presenting as an asymptomatic decrease in the left ventricular ejection fraction (LVEF), occurring in about one fourth of patients. It mainly occurs during the first three months of the treatment and leads to the treatment discontinuation in about 5% of patients [12]. In 14% of patients receiving trastuzumab, an increase in troponin I levels has been observed A multivariate analysis showed that observed troponin increase was an independent predictor of cardiotoxicity and in these patients LVEF did not recover. Therefore, troponin I elevation can identify patients who are at risk of developing cardiac disfunction that might not be recovered [13].

Little is known about the cardiotoxicity of pertuzumab and trastuzumab emtansine. Pertuzumab is a recombinant humanised monoclonal antibody targeting HER. It is always given in combination with trastuzumab. Its cardiotoxic effect is far less known. An APHINITY trial evaluated the effectiveness and safety of pertuzumab when added to combination of chemotherapy and trastuzumab (dual blockade) in the adjuvant treatment of breast cancer patients. After 6 years of follow up, the incidence of primary cardiac events was less than 1% in both groups of patients. No new safety signals regarding the cardiotoxicity of dual blockade have been detected [14,15]. However, in a meta-analysis of eight randomised controlled trials that included 8420 patients, pertuzumab was associated with an almost two-fold increased risk of heart failure. No other cardiotoxic effects were observed [16].

T-DM1 is a combination of trastuzumab, a HER2 antibody, and emtansine, which is an anti-microtubule agent. Data on its cardiotoxicity are scarce. It was compared to trastuzumab in a KATHERINE trial that enrolled 1486 patients with HER2 breast cancer that did not achieve a complete pathological response on primary systemic therapy. After a median follow up of 41 months, the cardiac events rate was 0.6% in patients receiving trastuzumab and 0,1% in patients receiving T-DM1 [17–19]. In a pooled analysis of seven

trials including over 1900 patients receiving T-DM1, the cardiac event rate was about 3%. This included congestive heart failure, LVEF drop, cardiac arrhythmias and cardiac ischemia. In almost 80% of patients, the events resolved upon treatment discontinuation [20]

According to the current position statement of the Cardio-Oncology Study Group of the Heart Failure Association and the Cardio-Oncology Council of the European Society of Cardiology, in patients with breast cancer that should receive anti-HER2 cancer therapies, a baseline cardiovascular risk assessment should be performed, based on previous cardiovascular disease (heart failure or cardiomyopathy, myocardial infarction or CABG, severe valvular heart disease, LVEF value, arrhythmias, stabile angina), cardiac biomarkers (troponin, BNP or NT-proBNP), demographic and cardiovascular risk factors (age, hypertension, diabetes mellitus, chronic kidney disease), current cancer treatment regimen (including anthracyclines before HER2-targeted therapy), previous cardiotoxic cancer treatment (including anthracyclines and chest radiotherapy) and lifestyle risk factors (smoking, obesity) [21].

In patients with breast cancer receiving both anthracyclines and trastuzumab, a measurement of troponin is recommended at baseline, before the commencement of trastuzumab-based therapy and after every four/three/two cycles of trastuzumab according to baseline cardiovascular risk assessment (low/medium/high) [7].

It is not clear whether a heart exposed to trastuzumab is more prone to radiation damage, since the data are equivocal [22,23]. In a retrospective trial of patients undergoing trastuzumab therapy in parallel with radiotherapy of either the right or left breast, radiation doses on the right ventricle (RV), LV and LAD were evaluated. Patients with left-sided breast cancer more often had arrhythmias and cardiac ischemia compared to patients with cancer of the right breast. Also, radiation dose on the RV, LV and LAD was positively correlated with LVEF decline [24].

In this study, we have enrolled patients with HER2-positive left breast cancer undergoing hypofractionated adjuvant left-breast radiotherapy concomitantly with anti-HER2 therapy: trastuzumab, a combination of trastuzumab and pertuzumab or trastuzumab emtansine (T-DM1). Values of high-sensitivity cardiac troponin I (hscTnI), as an early cardiotoxicity biomarker, have been measured prior to and upon completion of radiotherapy. Dose volume histograms (DVH) were generated for cardiac structures and correlated with the increment of hscTnI values in order to define acceptable radiation doses on the heart, left ventricle (LV) and left anterior descending artery (LAD).

## 2. Materials and Methods

### 2.1. Patient Population

In this single centre, cohort, prospective, observational trial, 61 female patients with HER2-positive early stage left breast cancer were enrolled. Patients underwent either breast-conserving surgery or mastectomy with axillary lymph node dissection or sentinel lymph node biopsy. Patients were treated with forward intensity-modulated radiotherapy. Clinical target volume consisted of the left breast or chest wall, with or without axillary or supraclavicular lymph nodes. All patients were receiving anti-HER2 therapy: trastuzumab, a combination of trastuzumab and pertuzumab or trastuzumab emtansine (T-DM1) concomitantly with irradiation. Exclusion criteria were myocardial infarction, symptomatic heart failure, chronic atrial fibrillation, malignant cardiac arrhythmias, pacemaker therapy, pulmonary embolism and renal failure.

The study was approved by the Institutional Ethics Committee and all patients signed their informed consent prior to recruitment. It was conducted from January 2022 to April 2023.

*2.2. Radiation Therapy*

Patients underwent 3D computer tomography treatment simulation during free breathing. Patients were placed on breast board in supine position with arms above their heads. CT slices were 2 mm thick and no intravenous contrast was used. Clinical target volume (CTV) consisted of left breast in 46 patients and left thoracic wall in 15 patients. In 1 patient, both breasts were irradiated. In 31 patients, regional lymph nodes were included in CTV, supraclavicular lymph nodes in 19 patients and both supraclavicular and axillary lymph nodes in 12 patients. Planning target volume (PTV) was created by adding 1 cm margin to account for intra- and inter-fraction movement. DVHs were generated for target volumes, lungs, spinal cord, heart, LV and LAD. In order to avoid interobserver variability, the same radiation oncologist contoured all structures (KA).

Radiation technique was forward intensity-modulated radiotherapy (fIMRT, field-in-field technique). Prescribed dose was 40.05 Gy in 15 fractions of 2.67 Gy over 3 weeks. Patients were irradiated during free breathing. Patient positioning was controlled using electronic portal-imaging device (EPID) prior to first five fractions and, thereafter, prior to every other fraction.

*2.3. High-Sensitivity Cardiac Troponin I Analysis*

HscTnI was analysed using Architect STAT Troponin I immunoassay (Abbott Laboratories, Abbott Ireland, Longford, Ireland). Serum samples were taken immediately before the first radiation fraction and immediately after the last radiation fraction. All samples were taken in the morning to avoid the influence of possible diurnal changes on hscTnI values. Patient samples were collected into CAT Serum Sep Clot Activator Vacuette with separator gel (Greiner Bio-One) and processed within 2 h of collection. The samples were centrifuged at $3000 \times g$ for 10 min. Analysis were performed on the Abbott Architect i2000 using reagents, calibrators and controls of the same manufacturer. Lowest detection limit was 1 ng/L.

Clinically significant increase was defined as a second value > 30% from the baseline and higher than 4 ng/L. Upon data completion, patients were divided in two groups: Group 1 with clinically significant hscTnI increase and Group 2 without clinically significant hscTnI increase.

*2.4. Statistical Analysis*

Quantitative data distribution normality was tested using Kolmogorov–Smirnov test. Qualitative features distribution was shown in contingency tables and differences in distribution were analysed using Fisher's exact test. Data were expressed as arithmetic means and standard deviation for normally distributed variables and as medians with interquartile range (IQR) for variables with significant deviation from normal distribution. Differences in distribution of numerical variables were analysed using Mann–Whitney U test and Wilcoxon test. The ROC curve was used to determine the optimal threshold value. Correlation of radiation doses and hscTnI increase was analysed using Spearman's rank correlation test. Data are shown as tables and figures. All statistical analyses are interpreted on a significance level of 5%.

## 3. Results

*3.1. Patients' Characteristics*

In total, 61 patients were enrolled in this trial. Their characteristics are shown in Table 1. There was no difference between the groups regarding their age, menopausal status, baseline hscTnI values, frequency of anthracycline based therapy, time since the last anthracycline cycle application, cardiac therapy and ACE inhibitors therapy. Besides ACE inhibitors, cardiac therapy also included angiotensin II receptor blockers, beta blockers, calcium channel blockers, imidazoline receptor agonists, diuretics and anti-aggregation agents.

During radiotherapy, 72% of all patients received hormonal therapy, 76.5% in the group with hscTnI increase and 70% in the group without hscTnI increase (no difference between the two groups, *p* = 0.1811). Within the group of patients receiving hormonal therapy, 70% were taking aromatase inhibitor (AI, either anastrozole or letrozole), 25% were receiving tamoxifen and 5% a combination of LHRH agonist goserelin and tamoxifen. No difference between the study groups regarding frequency or type of hormonal treatment has been observed.

In 6 patients in Group 1 (35.4%) and 23 patients in Group 2, the clinical target volume consisted of breast only. In 10 patients in Group 1 (58.8%) and 21 patients in Group 2 (47.7%), regional lymph nodes were also involved in CTV. In one patient in Group 1, both left and right breast were irradiated. There was no statistically significant difference between the groups in terms of clinical target volume comprehensiveness.

Data are shown in Table 1.

**Table 1.** Patients' and treatments' characteristics.

| | All N = 61 | Group 1 N = 17 | Group 2 N = 44 | *p*-Value |
|---|---|---|---|---|
| Age (x +/− SD) | 58 ± 11 | 55 ± 12.3 | 59 ± 10.5 | 0.2767 |
| Premenopausal | 16 (26%) | 7 (41%) | 9 (20%) | 0.1018 |
| hscTnI (ng/L) baseline (M,IQR) | 4 (2–7) | 5 (3–7) | 4 (2–7) | 0.8336 |
| Anthracycline use | 30 (49%) | 11 (65%) | 19 (43%) | 0.1811 |
| Time between anthracycline and RT in days (M,IQR) | 208.5 (188–227) | 216 (190–245) | 208 (185–219) | 0.3015 |
| Hormonal therapy | 44 (72%) | 13 (76.5%) | 31 (70%) | 0.1811 |
| Tamoxifen | 11 (25%) | 4 (30.8%) | 7 (22.6%) | 0.7221 |
| AI (anastrozole, letrozole) | 31 (70%) | 8 (61.5%) | 23 (74.2%) | 0.7979 |
| Goserelin + tamoxifen | 2 (5%) | 1 (7.7%) | 1 (3.2%) | 0.5208 |
| Clinical target volume | | | | |
| Left breast | 29 (47.5%) | 6 (35.4%) | 23 (52.3%) | 0.6069 |
| Left breast/thoracic wall + lymph nodes | 31 (51%) | 10 (58.8%) | 21 (47.7%) | 0.8099 |
| Both breasts | 1 (1.55%) | 1 (5.8%) | 0 | - |
| Cardiac therapy | 26 (43%) | 8 (47%) | 18 (41%) | 0.6658 |
| ACE inhibitors | 17/61 (28%) | 5/17 (29%) | 12/44 (27%) | 0.8684 |

*3.2. Anti-HER2 Treatments' Characteristics*

Before the commencement of radiotherapy, patients received either trastuzumab alone (28%), a combination of trastuzumab and pertuzumab (49%), T-DM1 (1%) or combination of trastuzumab and pertuzumab followed by T-DM1 (21%). No difference between the two groups has been shown regarding the type of anti-HER2 regimen or number of cycles of anti-HER2 therapy prior to radiotherapy.

During radiotherapy, patients received either trastuzumab alone (31%), a combination of trastuzumab and pertuzumab (46%) or T-DM1 (23%). Again, no difference between the two groups has been shown regarding the frequency of any anti-HER2 regimen. There was no difference observed between the two groups in terms of the day (fraction) of radiotherapy treatment on which anti-HER2 therapy was administered.

Data are shown in Table 2.

**Table 2.** Anti-HER2 therapy before and during radiotherapy.

| Anti-HER2 Therapy | All N = 61 | Group 1 N = 17 | Group 2 N = 44 | *p*-Value |
|---|---|---|---|---|
| Before radiotherapy | | | | |
| Trastuzumab | 17 (28%) | 4 (23%) | 13 (30%) | 1.0000 |
| Trastuzumab/pertuzumab | 30 (49%) | 9 (53%) | 21 (48%) | 1.0000 |
| T-DM1 | 1 (2%) | - | 1 (2%) | - |
| Trastuzumab/pertuzumab T-DM1 | 13 (21%) | 4 (23%) | 9 (20%) | 1.0000 |
| Number of cycles of anti-HER2 therapy before RT (M,IQR) | 7 (5–8) | 7 (5–8.25) | 7 (5.25–8) | 0.9934 |
| During radiotherapy | | | | |
| Trastuzumab | 19 (31%) | 4 (23.5%) | 15 (34%) | 0.7665 |
| Trastuzumab/pertuzumab | 28 (46%) | 9 (53%) | 19 (43%) | 0.8024 |
| T-DM1 | 14 (23%) | 4 (23.5%) | 10 (23%) | 1.0000 |
| RT fraction with anti-HER2 therapy application (M,IQR) | 9 (5–12) | 10 (4.75–12) | 8 (5.5–12) | 0.8590 |

*3.3. High-Sensitivity Cardiac Troponin I Values*

For the whole study population, the median (IQR) hscTnI values were 4 (2–7) ng/L before radiotherapy and 5 (3–10) ng/L after radiotherapy. A clinically significant increase in hscTnI, defined as a second value > 30% from baseline and higher than 4 ng/L, occurred in 17 patients (Group 1). The median values (IQR) were 4 (2–7) ng/L before RT and 8 (5–11) ng/L after RT. For Group 2, the baseline values were 4 (2–7) ng/L and 3 (2–6) ng/L upon treatment completion (Table 3). The values are shown in Table 1. Data are graphically presented in Figure 1.

**Table 3.** High-sensitivity cardiac troponin I values.

| | All N = 61 | Group 1 N = 17 | Group 2 N = 44 | *p*-Value |
|---|---|---|---|---|
| hscTnI (ng/L) baseline (M,IQR) | 4 (2–7) | 5 (3–7) | 4 (2–7) | 0.8336 |
| hscTnI (ng/L) after RT (M,IQR) | 5 (3–10) | 8 (5–11) | 3 (2–6) | 0.0053 |

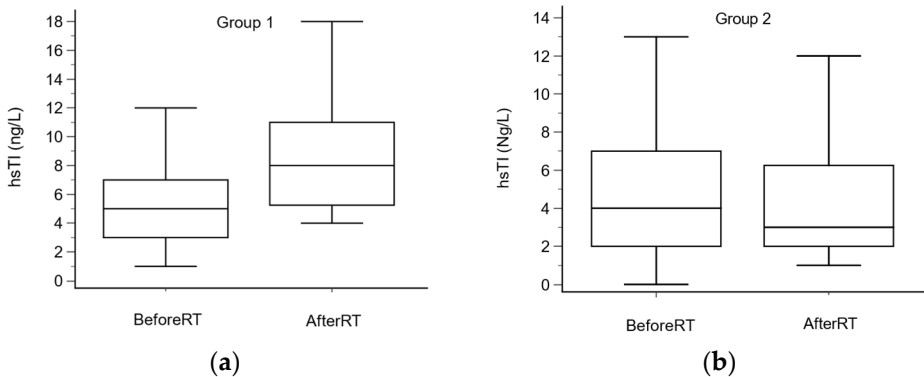

**Figure 1.** High-sensitivity cardiac troponin I values. Box plot figures to describe hscTnI before and after radiotherapy in (**a**) patients with increase (Group 1, 17 pts) and (**b**) in patients without an increase (Group 2, 44 pts). Borders of the box present Q1 and Q3, middle lines present medians and error bars above and below present maximums and minimums.

*3.4. Cardiac Doses*

The dose volume histograms for heart, left ventricle (LV) and left anterior descending artery (LAD) for Group 1 and Group 2 are shown in Figure 2. The cardiac doses for

both groups are shown in Table 4. Since the risk of cardiac radiation damage correlates with the volume of the cardiac structure receiving a certain radiation dose, besides mean and maximal doses, we have also selected 10 dose volume points for each cardiac structure. For all observed structures, the mean radiation doses were significantly higher in the Group 1 patients with hscTnI increase ($p = 0.02$ for heart, $p = 0.03$ for LV and $p = 0.04$ for LAD). A statistically significant difference between the groups was observed for AUC for heart and left ventricle ($p = 0.01$ for both), for volume of heart receiving 2 Gy and 4 Gy radiation dose ($p = 0.01$ and $p = 0.02$, respectively) and for volume of left ventricle receiving 2 Gy, 4 Gy and 38 Gy radiation doses ($p = 0.01$, $p = 0.02$ and $p = 0.03$, respectively), all values being higher for Group 1. Also, in Group 1, larger volumes of LAD received 2 Gy, 4 Gy and 5 Gy radiation doses ($p = 0.03$, $p = 0.02$ and $p = 0.02$, respectively), compared to Group 2. In conclusion, in the patients in Group 1 who had an increase in hscTnI levels, larger volumes of the heart, LV and LAD received low radiation doses than in patients in Group 2 (patients without hscTnI increase).

Dose volume constraints for hscTnI increase are shown in Table 5.

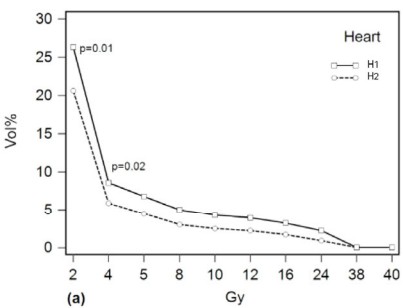

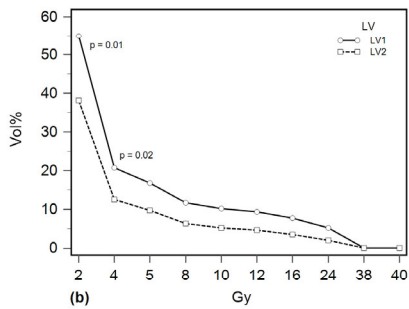

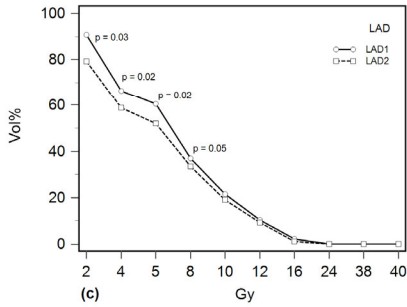

**Figure 2.** DVH curves for (**a**) heart, (**b**) left ventricle (LV) and (**c**) left anterior descending artery (LAD) in patients with high-sensitivity cardiac troponin I (hscTnI) increase (Group 1, 17 pts) and in patients without hscTnI increase (Group 2, 44 pts).

**Table 4.** Radiation doses on heart, left ventricle (LV) and left anterior descending artery (LAD) in patients with high-sensitivity cardiac troponin I (hscTnI) increase (Group 1, 17 pts) and in patients without hscTnI increase (Group 2, 44 pts).

| Cardiac Structure | Group 1 (N = 17) Median (IQR) | Group 2 (N = 44) Median (IQR) | *p* Mann–Whitney U |
|---|---|---|---|
| **Heart** | | | |
| Dmean (Gy) | 2.4 (1.9–3.9) | 1.8 (1.5–2.5) | 0.02 |
| Dmax (Gy) | 38.3 (35.3–39.6) | 37.4 (33.8–38.6) | 0.22 |
| V2 (%) | 26.3 (21.1–35.2) | 20.5 (18.9–23.6) | 0.01 |
| V4 (%) | 8.6 (6–16) | 5.8 (4.4–8.9) | 0.02 |
| V5 (%) | 6.8 (4–13.8) | 4.5 (3.2–7.1) | 0.07 |
| V8 (%) | 5 (2.3–10.7) | 3 (1.5–5) | 0.08 |
| V10 (%) | 4.3 (1.7–9.5) | 2.5 (1–4.2) | 0.08 |
| V12 (%) | 3.9 (1.3–8.5) | 2.1 (0.8–3.7) | 0.08 |
| V16 (%) | 3.2 (0.8–7) | 1.6 (0.5–3) | 0.09 |
| V24 (%) | 2.2 (0.3–4.4) | 0.8 (0.1–4.7) | 0.10 |
| V38 (%) | 0 (0–0.3) | 0 (0–0) | 0.27 |
| V40 (%) | 0 (0–0.3) | 0 (0–0) | 0.74 |
| AUC (%) | 72.9 (72.1–74.3) | 51.4 (51.4–54.3) | 0.01 |
| **LV** | | | |
| Dmean (Gy) | 4.7 (2.8–6.1) | 2.9 (2.3–4.2) | 0.03 |
| Dmax (Gy) | 38.4 (35.3–39.6) | 37.2 (33.5–38.5) | 0.13 |
| V2 (%) | 54.9 (41.3–62.8) | 34.8 (32.3–48.3) | 0.01 |
| V4 (%) | 20.8 (12.7–27.7) | 12.5 (10.4–16.1) | 0.02 |
| V5 (%) | 16.8 (9–22.8) | 9.7 (6.3–15.6) | 0.06 |
| V8 (%) | 11.7 (5.2–17.5) | 6.3 (3.2–10.8) | 0.08 |
| V10 (%) | 10.2 (3.9–15.7) | 5.2 (2.1–9.2) | 0.08 |
| V12 (%) | 9.3 (3.1–14.4) | 4.6 (16–8.1) | 0.08 |
| V16 (%) | 7.7 (1.8–12.3) | 3.7 (0.8–6.5) | 0.08 |
| V24 (%) | 5.2 (0.7–8.7) | 2 (0.2–4.2) | 0.09 |
| V38 (%) | 0.1 (0–0.6) | 0 (0–0) | 0.03 |
| V40 (%) | 0 (0–0) | 0 (0–0) | 0.8 |
| AUC (%) | 67.8 (67.1–72.9) | 51.4 (51.4–52.9) | 0.01 |
| **LAD** | | | |
| Dmean (Gy) | 6.8 (6.3–19.2) | 6.2 (5.1–6.9) | 0.04 |
| Dmax (Gy) | 22.4 (16–37.8) | 19.1 (17.2–25) | 0.21 |
| V2 (%) | 90.7 (82.2–99.9) | 79.2 (71.8–88.9) | 0.03 |
| V4 (%) | 66.3 (61.8–74.1) | 58.8 (48–67.8) | 0.02 |
| V5 (%) | 60.5 (55.6–70.6) | 52 (41.3–63) | 0.02 |
| V8 (%) | 37 (31.1–55.7) | 33.5 (16.3–38.4) | 0.05 |
| V10 (%) | 21.5 (16.1–53.3) | 19.1 (6.6–23.2) | 0.07 |
| V12 (%) | 10.4 (6.1–51.3) | 9.2 (2.5–16.1) | 0.08 |
| V16 (%) | 2.2 (0–48.2) | 1.1 (0.1–3.5) | 0.24 |
| V24 (%) | 0 (0–44.3) | 0 (0–0) | 0.16 |
| V38 (%) | 0 (0–0) | 0 (0–0) | 0.19 |
| V40 (%) | 0 (0–0) | 0 (0–0) | 0.54 |
| AUC (%) | 63.2 (58.6–68.6) | 60 (57.1–63.6) | 0.12 |

LV—left ventricle, LAD—left anterior descending artery, hscTnI—high sensitivity cardiac troponin I, Dmean—mean radiation dose to the structure, Dmax—maximal point radiation dose in the structure, V 40/38/24/16/12/10/8/5/4/2—the volume of structure receiving 40 Gy, 38 Gy, 24 Gy, 16 Gy, 12 Gy, 10 Gy, 8 Gy, 5 Gy, 4 Gy and 2 Gy doses, AUC—area under curve.

**Table 5.** Dose volume constraints for hscTnI increase (ROC curve data).

| Radiation Dose (Gy) | Heart Dose-Volume Constraint (%) | LV Dose-Volume Constraint (%) | LAD Dose-Volume Constraint (%) |
|---|---|---|---|
| 2 | >19.7 | >38.7 | >86.4 |
| 4 | >10.2 | >12.1 | >59.6 |
| 5 | >8.6 | >14.5 | >52.4 |
| 8 | >4.7 | >10 | >16.7 |
| 10 | >3.9 | >8.2 | >7.1 |
| 16 | >2.6 | >7.3 | >1.6 |
| 38 | >0.1 | >0.1 | >0 |
| 40 | >0.1 | >0.2 | ≤0 |

LV—left ventricle, LAD—left anterior descending artery, V 40/38/24/16/12/10/8/5/4/2—the volume of structure receiving 40 Gy, 38 Gy, 24 Gy, 16 Gy, 12 Gy, 10 Gy, 8 Gy, 5 Gy, 4 Gy and 2 Gy radiation dose. Dose-volume constraint—the threshold value for hscTnI value increase.

## 4. Discussion

Upon completion of left breast radiotherapy, hscTnI levels increased in about one fourth of patients and that increase was correlated with radiation doses on the heart and its structures, LV and LAD suggesting subclinical myocardial damage caused by irradiation. After lower radiation doses, such as those observed in our study, the underlying mechanism is microvasculature damage and inflammatory changes. They lead to focal ischemia resulting in myocyte damage and subsequent troponin release [8]. Based on data from the literature, troponin elevation during cancer treatment is correlated with the later development of cancer-treatment-related cardiac dysfunction that might not recover [10,13,25,26]. Although cut-off values of high-sensitivity cardiac troponin are yet to be defined, its evaluation during cancer treatment can identify patients that require a more thorough follow-up of cardiac function.

In our study, no difference between the groups has been observed regarding patients' age, menopausal status, prior cardiac therapy and the use of ACE inhibitors.

Anthracyclines are cytostatic antibiotics that have been in use in oncology since the 1960s. The risk for developing cardiotoxicity caused by anthracyclines is proportional to their cumulative dose; it occurs in up to 5% of patients with doses of 400 mg/m2. It can occur years after therapy with anthracyclines and its incidence rises with the time flow from the last application. It is more common in older patients, patients with previous heart conditions, in patients that underwent radiotherapy of the thorax and, in last two decades, in breast cancer patients receiving trastuzumab. The underlying mechanism of anthracycline-caused cardiotoxicity is still not clear. It presents with hypokinetic cardiomyopathy diagnosed by a decrease in left ventricle ejection fraction (LVEF) and eventually leads to heart failure. The early diagnosis and early onset of therapy with ACE inhibitors and beta blockers can result in LVEF improvement. If diagnosed at a later stage, anthracycline-caused cardiotoxicity is usually irreversible and has poor prognosis [27].

Anthracycline-based chemotherapy is known to elevate troponin levels. That elevation is transitional and correlates with the development of cardiotoxicity, including a decline in left ventricular ejection fraction [25,26]. In our study, one half of the patients received anthracycline-based chemotherapy; 65% in Group 1 and 43% in Group 2 (*p* = 0.1811). The median times from the application of the last cycle of anthracycline and the commencement of radiotherapy were 216 days for Group 1 (IQR: 190–245) and 208 days for Group 2 (IQR: 185–219), *p* = 0.3015. In conclusion, no statistically significant difference between the groups in terms of anthracycline use has been observed. Therefore, we did not attribute the hscTnI increase observed in Group 1 to previous anthracycline use.

Trastuzumab itself can also increase troponin levels. As with anthracyclines, troponin increment caused by trastuzumab is transitional and was shown to be predictive of

later cardiotoxicity [13]. In our study, no difference between the groups regarding number of anti-HER2 therapy applications prior to radiotherapy has been observed; in both groups, the median number of cycles was 7 (IQR: 5–8.25 for Group 1 and 5.25–8 for Group 2; *p* = 0.9934).

Also, there was no difference regarding the type of anti-HER2 therapy, either before or during radiotherapy. Prior to radiotherapy, patients were receiving trastuzumab, a combination of trastuzumab and pertuzumab, T-DM1 or combination of trastuzumab and pertuzumab followed by T-DM1. During radiotherapy, patients were given either trastuzumab, a combination of trastuzumab and pertuzumab or T-DM1. That is in concordance with most of the literature data showing no difference in cardiotoxicity regarding the type of anti-HER2 treatment [14,15,17–20]. One exception is the meta-analysis of eight randomised controlled trials revealing a higher risk of heart failure associated with pertuzumab [16]. However, although patients in these trials were irradiated concomitantly with anti-HER2 therapy, data on the radiation doses on cardiac structures are lacking.

To exclude the possible acute effect of the application of anti-HER2 therapy during radiotherapy on hscTnI release, we have recorded the radiation fraction with which anti-HER2 therapy was administered. Again, no difference between the groups has been observed. The median radiotherapy fraction values were 10 for Group 1 (IQR: 4.75–12) and 8 for Group 2 (IQR: 5.5–12), *p* = 0.859. Based on the abovementioned, we have excluded the effect of anti-HER2 therapy application on hscTnI levels.

In terms of hormonal therapy, this was prescribed in about three fourths of patients during radiotherapy. About 75% of the patients in each group that were receiving hormonal therapy were given aromatase inhibitor. No difference between the groups has been shown regarding either the use or type of hormonal therapy. Therefore, it is not likely that hormonal therapy might have influenced hscTnI release.

The clinical target volume was either left breast (in the case of breast-conserving surgery), with or without lymph nodes or left thoracic wall with lymph nodes. If included in CTV, lymph nodes were either supraclavicular or both supraclavicular and axillary lymph nodes. The target volumes were determined according to the current guidelines and based on the initial clinical stage of the disease, the type and extent of axillary surgery and the pathohistological report. In one patient in Group 1, both breasts were irradiated upon breast-conserving surgery due to bilateral breast cancer. Although the proportion of patients with lymph nodes included in the target volume was slightly higher in Group 1, 58.8%, compared to 47.7% in Group 2, this was not statistically significant.

The two groups differed only in radiation doses on cardiac structures. The mean heart dose median was 2.4 Gy for Group 1 (IQR: 1.9–3.9) vs. 1.8 Gy for Group 2 (IQR: 1.5–2.5). This is in accordance with the finding that the risk of major cardiac event is proportional to the mean dose to the whole heart and increases linearly by 7.4% per Gray [2]. When analysing DVHs, the measured dose volume points were different for low doses: V2 (volume of heart receiving 2 Gy) and V4 (volume of heart receiving 4 Gy), meaning that, in Group 1 with hscTnI increase, larger volumes of the heart were exposed to more radiation than in Group 2. Though it is still unclear what is more detrimental, a lower dose for a larger volume or a higher dose for a small volume, our data indicate that a low dose applied on a larger volume contributed to acute myocyte damage, resulting in subsequent troponin release.

For the left ventricle and left anterior descending artery, the findings were similar. There was a statistically significant difference for mean doses between the two groups. For LV, the medians were 4.7 for Group 1 (IQR: 2.8–6.1) and 2.9 Gy for Group 2 (IQR: 2.3–4.2). The two groups differed for V2 and V4, but also for V38, with all volumes being larger in Group 1.

LAD is the most anterior part of the heart situated near the clinical target volume. According to data from the literature, the mean dose on LAD higher than 5 Gy was con-

nected with an increased requirement of coronary intervention in LAD years after the completion of radiotherapy [5]. In our study, the median dose on LAD was higher than 5 Gy in both groups: it was 6.8 Gy for Group 1 (IQR: 6.3–19.2) and 6.2 Gy for Group 2 (IQR: 5.1–6.9). The measured dose volume points were different for low doses, with V2, V4 and V5, again, being higher in Group 1 with hscTnI increase.

Radiation-dose-dependent troponin release during irradiation of the left breast has already been described in the literature. High-sensitivity cardiac troponin T was measured prior to, during and after left breast radiotherapy. In patients with increased hscTnT values, higher radiation doses on the whole heart and LV were reported, as well as V15 and V20 for LAD [11]. However, the patients in this trial did not receive chemotherapy prior to irradiation or anti-HER2 therapy either prior to or during irradiation. Lymph nodes were not included in the target volume and, also, fractionation schemes were different than in our study; patients received either 50 Gy in 2 Gy daily fraction over 5 weeks or 42.56 Gy in 2.66 Gy fractions over 3.5 weeks. According to the institutional guidelines based on the data in the literature, the patients in our trial received 40.05 Gy in 2.67 Gy fractions over 3 weeks—a hypofractionated regimen [28,29]. Therefore, the applied radiation dose in our patients was somewhat lower and the overall treatment time was shorter.

Numerous trials did not show any difference in cardiotoxicity between the conventional radiotherapy (CFRT) and hypofractionated radiotherapy (HFRT). In START trials, schedules of 41.6 Gy or 39 Gy in 13 fractions over 5 weeks and 40 Gy in 15 fractions over 3 weeks were compared to the conventional fractionation scheme of 50 Gy in 25 fractions over 5 weeks. After 10 years of follow-up, there was no difference in the frequency of ischemic heart disease between the groups of patients that received HFRT and the patients irradiated with conventional fractionation [29]. Long-term data from a Canadian trial comparing 42.56 Gy in 16 fractions over 22 days with 50 Gy in 25 fractions over 5 weeks are similar; after 12 years of follow-up, few cardiac-related deaths were observed in both groups of patients. There was no increase in cardiac-related deaths in patients who were irradiated with a hypofractionated schedule [30]. In a meta-analysis that was published in 2020, the authors analysed the data from 25 clinical trials that enrolled 3871 postmastectomy patients and compared HFRT with CFRT in terms of both treatment efficacy and toxicity. No difference in late cardiac toxicity between the schedules was observed [31]. In an analysis of the data of 510 breast cancer patients irradiated between 2002 and 2006, either conventionally or with a hypofractionated schedule, cardiac toxicity was evaluated. The rate of ischaemic cardiac disease was low in both group of patients. According to the trial data, the fractionation schedule had no influence on the frequency of cardiotoxicity [32].

The strengths of our study are the clear inclusion criteria that reduce uncertainty and error factors, and the well-balanced subgroups of patients. The limitations could be the possibly confounding effect of anthracycline use, though groups were balanced in this regard and there was a variability of cardiac therapy.

Based on the abovementioned data, we have calculated dose-volume constraints for the whole heart, LV and LAD in order to define the radiation doses above which troponin release as a result of radiation damage can be expected. They should not be considered absolute safe cardiac doses for this patient population, but rather as guidance that might be considered for treatment planning. However, patients with troponin increase should be followed more carefully for the early diagnosis of cardiac dysfunction and timely implementation of cardioprotective treatment strategies in order to improve both oncological and cardiovascular outcomes. It is yet to be explored if this hscTnI increase can predict the future risk of cardiovascular morbidity and mortality in this group of patients.

## 5. Conclusions

In patients with HER2-positive breast cancer undergoing adjuvant hypofractionated left breast radiotherapy concomitantly with anti-HER2 therapy, high-sensitivity cardiac

troponin I is being released, dependent on the radiation dose on the heart and its structures. The results of this study can partially contribute to understanding early cardiotoxicity development and detection.

**Author Contributions:** Conceptualisation, K.A. and L.B.-O.; methodology, K.A., L.M. and L.B.-O.; formal analysis, K.A. and M.B.; investigation, K.A.; resources, L.M. and L.B.-O.; data curation, K.A. and M.B.; writing—original draft preparation, K.A.; writing—review and editing, L.M., M.B. and L.B.-O.; project administration, K.A. and L.B.-O. All authors have read and agreed to the published version of the manuscript.

**Funding:** This research received no external funding.

**Institutional Review Board Statement:** The study was conducted in accordance with the Declaration of Helsinki, and approved by the Institutional Ethics Committee of Sestre Milosrdnice University Hospital Center, Zagreb, Croatia (003-06/21-03/011, 8 April 2021).

**Informed Consent Statement:** Informed consent was obtained from all subjects involved in the study.

**Data Availability Statement:** The datasets analysed or generated during the current study are available from the corresponding author upon reasonable request.

**Conflicts of Interest:** The authors declare no conflict of interest.

## Abbreviations

| | |
|---|---|
| hscTnI | high-sensitivity cardiac troponin I |
| Gy | Gray |
| LN | lymph nodes |
| ROC | receiver operating characteristics |
| RT | radiation therapy |
| DVH | dose volume histograms |
| LAD | left anterior descending artery |
| LV | left ventricle |
| LVEF | left ventricular ejection fraction |
| T-DM1 | trastuzumab emtansine |
| CFRT | conventional fractionated radiotherapy |
| HFRT | hypofractionated radiotherapy |

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
