# Peer review of "Correlation of High-Sensitivity Cardiac Troponin I Values and Cardiac Radiation Doses in Patients with Left-Sided Breast Cancer Undergoing Hypofractionated Adjuvant Radiotherapy with Concurrent Anti-HER2 Therapy"

_curroncol, doi:10.3390/curroncol30100654_

Round 1
Reviewer 1 Report
the study relates to cardiotoxicity of her2 directed therapy and radiotherapy - a very common and highly relevant scenario- in a cohort of over 60 patients the authors identify this issue with a sensitive blood test elevation which is then correlated with radiation fields - the clinical rationale is logical and would suggest careful cardiooncology assessment in at risk patients where treatment planning identifies cardiac exposure.
the methods are clearly explained however no explanation is given of cardiac therapy and what criteria were used to identify this (Table 1). The results section has over 68 tests for statistical significance despite only 61 subjects in the study - no reference is made of adjustment for multiple comparissons by the authors
here are multiple examples in the paper eg line 41 and 42 where prepositions are not included eg reduces risk should be reduces the risk ;
line 52 connected should be changed to associated
Author Response
Please, see the attachment

Reviewer 2 Report
In the article "Correlation of High Sensitivity Cardiac Troponin I Values and Cardiac Radiation Doses in Patients with Left-sided Breast Cancer Undergoing Hypofractionated Adjuvant Radiotherapy with Concurrent Anti-HER2 Therapy", Antunac et al. choose a novel method to demonstrate the cardiac toxicity of radiotherapy by evaluating the values in the dynamics of high sensitivity cardiac troponin 1. I greatly appreciate the choice of a balanced group of patients in the two subgroups, but especially the inclusion criteria (HER2 + cases, moderate hypofractionated radiotherapy on breast only) that reduce uncertainty and error factors. The idea is interesting and the study is well structured, demonstrating the involvement of radiotherapy of the left breast as a decisive factor in the dynamics of the biomarker. However, I would consider useful the summary presentation of the cardiac toxicity profile of anthracyclines and anti-HER2 biological therapies, as well as the current data on hypofractionated radiotherapy in relation to standard radiotherapy in cardiac toxicity. I would also suggest mentioning some data about cardiac sub-structures and the dose-volume-toxicity relationship. I believe that these mentions would make the work of wider interest both for the cardiologist and for the medical oncologist or radiation oncologist. I also appreciate the remark related to the possible detrimental effect of small doses of radiation in large volumes - more specific to techniques with modulated intensity and especially in the VMAT technique. For this reason, reporting to data from the literature (data in which the long-term follow-up includes cases treated with opposite tangential beams) is difficult. Even the absence in the historical studies of TDM-1 and Pertuzumab makes it necessary to re-evaluate the data in the current context and for this reason the study must be appreciated.
Author Response
Please, see the attachment

Reviewer 3 Report
MS: Correlation of High Sensitivity Cardiac Troponin I Values and Cardiac Radiation Doses in Patients with Left-sided Breast Cancer Undergoing Hypofractionated Adjuvant Radiotherapy with Concurrent Anti-HER2 Therapy
The recent advances in the area of breast cancer (BC) treatment have contributed to an improvement in the prognosis of many patients with BC. However, cardiotoxicity caused by such treatments often deteriorates patient outcomes. Adverse events (AEs) affecting the cardiovascular (CV) system are often related to chemotherapy (CHT) agents (e.g., anthracyclines), targeted therapies, immunotherapies (e.g., anti-HER2 therapies), and radiotherapy (RT).
This study attempts to assess the impact of radiation dose on cardiac structures, based on the values of cardiotoxicity marker high sensitivity cardiac troponin I (hscTnI) in patients with HER2 positive left BC, who are receiving adjuvant concomitant anti-HER2 therapy and RT, and to establish the correlation of hscTnI values with cardiac radiation doses. According to the study findings, in patients with HER2-positive BC receiving such therapies, hscTnI has been released, depending on the RT dose on the heart and its structures (LV and LAD). This may explain some aspects of cardiotoxicity in this group of patients. However, further studies are necessary to investigate whether the hscTnI increase can predict future CV morbidity and mortality risk in this patient population.
Some suggestions to be considered by the Authors for the revision are provided below.
In the Discussion section, the Authors could elaborate on the statement: “Patients with troponin increase should be followed more carefully for early diagnostic of cardiac dysfunction and timely implementation of cardioprotective treatment strategies in order to improve both oncological and cardiovascular outcomes”. How the patients with elevated hscTnI should be followed? What diagnostic tests should be checked (e.g., other CV biomarkers, like BNP ( B‐type natriuretic peptide), or NT‐proBNP (N‐terminal pro‐B‐type) and how often? Which ‘cardioprotective treatment strategies’ should be implemented (e.g., ACEIs or ARBs, BBs, etc.)? When exactly (e.g., prior to the planned cardiotoxic anticancer therapies or during these therapies) and by whom (e.g., cardiologist, cardio-oncologist, internist, family physician, etc.) should these strategies be implemented? How to reinforce the patients' adherence/engagement in these cardioprotective therapies “ to improve both oncological and CV outcomes” ? (e.g., by patient education, close communication with treatment teams, active changing of the modifiable CV risk factors/CVD such as moderate exercise and healthy nutrition). Also, in clinical trials that investigate anticancer therapies with potential CV AEs, cardio‐oncologists should design the monitoring procedures focused on CV safety. Perhaps, the Authors could briefly mention the selection of high-CV-risk patients, who should be more carefully monitored, while receiving effective treatments for BC.
The Authors should add the study strengths and limitations section.
Author Response
Please, see the attachment
